# Diagnosis and Management of Functional Tic-Like Phenomena

**DOI:** 10.3390/jcm11216470

**Published:** 2022-10-31

**Authors:** Irene A. Malaty, Seonaid Anderson, Shannon M. Bennett, Cathy L. Budman, Barbara J. Coffey, Keith A. Coffman, Erica Greenberg, Joseph F. McGuire, Kirsten R. Müller-Vahl, Michael S. Okun, Julio Quezada, Amy Robichaux-Viehoever, Kevin J. Black

**Affiliations:** 1Department of Neurology, Norman Fixel Institute for Neurological Diseases, University of Florida College of Medicine, Gainesville, FL 32608, USA; 2Neuro-Diverse.org, Brussels, Belgium; 3Department of Psychiatry, Weill Cornell Medicine/NewYork-Presbyterian, New York, NY 10065, USA; 4Department of Psychiatry, Northwell Health, Zucker School of Medicine, Hofstra/Northwell, Uniondale, NY 11549, USA; 5Department of Psychiatry and Behavioral Sciences, University of Miami Miller School of Medicine, Miami, FL 33136, USA; 6Children’s Mercy Hospital, University of Missouri-Kansas City School of Medicine, Kansas City, MO 64108, USA; 7Department of Psychiatry, Massachusetts General Hospital, Boston, MA 02129, USA; 8Department of Psychiatry and Behavioral Sciences, Johns Hopkins University School of Medicine, Baltimore, MD 21205, USA; 9Department of Psychiatry, Social Psychiatry and Psychotherapy, Hannover Medical School (MHH), 30625 Hannover, Germany; 10Department of Neurology, Washington University in St. Louis, St. Louis, MO 63130, USA; 11Departments of Psychiatry, Neurology, Radiology and Neuroscience, Washington University in St. Louis, 660 S. Euclid Ave., St. Louis, MO 63110, USA

**Keywords:** Tourette syndrome, functional neurological disorder, functional tics, functional tic-like behaviors, diagnosis, management

## Abstract

Over the past 3 years, a global phenomenon has emerged characterized by the sudden onset and frequently rapid escalation of tics and tic-like movements and phonations. These symptoms have occurred not only in youth known to have tics or Tourette syndrome (TS), but also, and more notably, in youth with no prior history of tics. The Tourette Association of America (TAA) convened an international, multidisciplinary working group to better understand this apparent presentation of functional neurological disorder (FND) and its relationship to TS. Here, we review and summarize the literature relevant to distinguish the two, with recommendations to clinicians for diagnosis and management. Finally, we highlight areas for future emphasis and research.

## 1. Background

### Tics and Tic-Like Behaviors

Tics are characterized by changing severity and semiology over time, which understandably can present diagnostic challenges for clinicians. Although there are no pathognomonic diagnostic signs or lab tests to confirm tics, much is known about the typical course and symptoms, and clinicians with expertise in tic disorders can recognize consistent patterns [1]. Tics are usually paroxysmal, occur in “bouts”, change in type and location, and may vary across individuals and even within an individual over time. Tics can be unusual in form and may be suggestible (i.e., increase with discussion of or attention to tics). Tics may improve with intense focus or concentration and worsen during times of personal or social excitement or stress. The COVID-19 pandemic has been one such period of heightened distress for many individuals; exacerbation of symptoms in those with existing tic disorders has been reported [2,3,4]. Interestingly, some children with tics experienced improvement in response to the decreased social exposures associated with remote school and work. At present, there is no consensus regarding the net impact of the pandemic on existing tics.

However, over the past 3 years, tic specialists worldwide have observed an increase in young people presenting for evaluation of tics and Tourette syndrome (TS) [5,6]. Some of these cases appear to represent an exacerbation of tics in people with primary tic disorders, whether the diagnosis was known to them previously or not. Others, however, present with distinct characteristics that differ significantly from those of typical tic disorders [5,7,8]. These tic-like behaviors manifest abruptly in adolescents without a prior history or family history of tics, and phenomenologically do not conform with typical clinical features associated with tics.

Cases of such tic-like behaviors have been described previously. Some may emerge in isolated clusters or endemically, and appear consistent with a social contagion model, as exemplified by an outbreak near Rochester, NY, that occurred a decade ago [9,10,11]. In general, however, the presentations of similar tic-like behaviors appeared relatively less frequently prior to 2019–2020 [12], and increased significantly at many medical centers in 2020–2021 [6,13,14]. Over the past decade, with the increase in access to and utilization of activities involving social media (e.g., TikTok, Instagram), tics and tic-like symptoms have gained greater visibility and interest. Some of those experiencing a sudden onset of tic-like behaviors have symptoms that appear very similar to presentations that were viewed on social media [15,16,17,18]. Given the reports of increased time spent on social media and the Internet during the pandemic, including by youth with established tic disorders [19], exposure to such content was amplified. There has been some discussion amongst researchers and clinicians as to the amplitude of the effects of social media and whether this has been over-emphasized [20,21,22].

As McGuire and colleagues wrote, “Spikes in atypical presentations of tic-like movements [and vocalizations] are not new and will likely recur. Thus, characterizing the phenomenology, investigating mechanisms underlying symptom onset, tracing the symptom trajectory to understand long-term outcomes, and developing effective treatments is of increasing importance” [23]. The Tourette Association of America (TAA) commissioned this article reviewing what is known about these cases and how to distinguish them from primary tic disorders including TS.

## 2. Understanding Tourette Syndrome

TS is a neurodevelopmental disorder that is characterized by motor and vocal tics and usually begins in early childhood or adolescence (modal onset 4–8 years of age [24,25,26,27,28,29,30]. TS is part of a spectrum of tic disorders, including persistent (chronic) motor or vocal tic disorder (more than 1 year) and provisional tic disorder (less than 1 year in duration).

TS often continues into early adolescence, and in some cases, adulthood. There is a sex ratio of about 3–4:1 boys to girls, though this ratio varies widely in available studies; and there is suggestion that tics subside earlier in men than in women, producing ratios nearer unity in adults [31]. Delay in diagnosis, or under- or misdiagnosis in females may exist, and better epidemiological studies that examine and compare typical phenomenology and natural history in males and females are needed. There is some evidence that suggests tics in females may be more complex, may begin later in life, and may be more often complicated by mood and anxiety disorders [31,32], but conflicting data also exist, and these points require further investigation [26]. Garris and Quigg (2021) argue that tics may begin later in females than in males, although the 3 studies they cite to support a difference are far from conclusive [31]. One of these reports retrospectively recalled onset in 11 females compared to 42 males, but provided no group mean ages of onset [29]. A second study (n = 148) reported a later onset age in girls, but the mean difference was only 1.2 years [25]. Most studies, including the largest one [33], found no significant sex difference in age of onset [34,35,36,37,38]. The only large, prospective study of youth beginning *before* tic onset found male predominance [30]. In summary, later onset in females is possible but unproven, and even the studies that reported a difference in age of onset found small differences, with mean onset by age 10. A recent blinded video review of 201 individuals found a statistically higher frequency of complex tics in males than in females [38].

Typically, the first experience of tics is a simple motor tic, such as exaggerated blinking [1]. Simple motor tics often begin in the head, face, or neck, with later tics affecting other body parts (rostro-caudal progression). On average, simple vocal (phonic) tics begin 2–3 years after the first motor tics. More complex tics may develop as well, usually gradually and over time. Coprophenomena (vulgar or profane words or actions) occur in a minority of individuals with TS, and tend to start about 5–6 years after tic onset [26,39,40]. Many individuals who experience tics, particularly those older than 10 years of age, describe a sensation or impulse (called a “premonitory urge”) just before the tic, that either is relieved or lessened in intensity for a short period of time after the tic. Most people with tics can suppress tics briefly with effort [21]. Finally, people with tics often have a close relative who also has a history of tics, obsessive compulsive disorder (OCD), or attention-deficit/hyperactivity disorder (ADHD), though family history is variable [41].

## 3. Understanding Functional Neurological Disorder

In functional neurological disorder (FND), a patient experiences neurological symptoms that cannot be attributed to a lesion or medical condition, but that are genuinely experienced and may cause distress. FND often prompts a search for associated psychological distress or history of trauma, though a compelling link between stressors and symptom onset may be absent [42]. Historically, understanding FND has been challenging. Ancient civilizations used the word “hysteria” due to the erroneous theory that the symptoms were the result of uterine dysfunction (or a wandering uterus) [43]. In the early 20th century, Briquet, Janet, Breuer, Charcot, and Freud recognized the role of attention and physical or emotional triggers in the genesis of functional symptoms, and pointed out the important difference between FND and feigned symptoms. Freud and Breuer suggested “that there was a natural excess of excitation which increased during and after puberty… the excess excitation triggered by emotional events is ‘converted’ into somatic phenomena” [43]. This theory in which emotional trauma was the central cause of FND gave rise to the term “conversion disorder” and prevailed until the late 20th century [44].

In the mid-20th century, single putatively functional symptoms often proved on follow-up to represent neurological or systemic disease [45,46]. This result led the Washington University criteria and their descendants to focus on lifelong, multiple somatizations as a reliable and valid diagnosis [47,48,49]. Unfortunately, this subgroup of patients represented the minority of patients with medically unexplained neurological symptoms. By the end of the 20th century, it became clear that with modern diagnostic methods, misdiagnosis of known neurological diseases as hysterical occurred only in 0.4% to 4% of patients seen by neurologists [50,51,52]. This result led to a renascence of interest in these phenomena, and a focus on requiring positive signs for diagnosis (“rule in”) rather than on the absence of evidence for better-understood, typical diagnoses (“rule out”) [53]. Former labels were abandoned based on patient preference in favor of the term “functional” neurological symptoms [54,55], or FND. 

Renewed interest in FND and its neurobiology have led researchers to discover multiple neurophysiological differences through brain imaging studies. While an in-depth review of these differences is beyond the scope of this review, we highlight some of the notable studies and their findings [43]: There are differences in neural activity during functional symptoms vs. feigned symptoms [54,56], abnormal neural perception or processing of sensory information [57], absent sense of agency of movements [58], and abnormal limbic and paralimbic activity [59,60] (Aybek et al., 2015; Aybek et al., 2014). These studies have contributed to the current understanding of FND as a neuropsychiatric illness with both neurophysiologic differences and psychological manifestations.

FND may occur in isolation or co-occur with better-understood movement disorders such as TS. This is well-established in other disorders, such as epilepsy, where a subset of individuals may experience paroxysmal non-epileptic seizures, or seizure-like spells not associated with any abnormal EEG activity [61]. Patients with functional movements of differing phenomenology may share some common characteristics and challenges [62].

Tic-like presentations due to FND and typical tics (as seen in TS) share some features, including suggestibility, distractibility, and worsening in times of stress [63]. Both conditions can manifest echophenomena (repetitions of external movements or sounds) [64]. Differential diagnosis of TS from functional tic-like presentations can therefore be challenging, especially for the general public, and even the general medical practitioner or clinical psychologist, and often requires subspecialty expertise and experience. However, some of the recent symptoms seen on social media and presenting in clinics clearly differ in some respects from previous reports of functional tic-like behaviors (discussed in the following sections). Importantly, any type of repetitive movement condition can be problematic, and deserves care, but making the correct diagnosis is critical both to avoid inappropriate interventions and to find the optimal treatment for the patient.

## 4. Typical FND Tic-Like Symptoms

The growing number of people with sudden onset tic-like phenomena typically do not manifest the usual tic symptoms and patterns. Instead, they often demonstrate the sudden onset of complex motor and vocal tic-like phenomena that differ substantially from those typically seen in youth with TS. 

Functional tic-like symptoms seen in recent years primarily occur in adolescents, and females appear to be at higher risk. They usually do not have an identified or family history of tics, though one needs to be thoughtful about a potential history of tics that went undiagnosed [5].

These individuals often have been experiencing co-occurring anxiety or depression and significant psychosocial stressors. Interestingly, there is preliminary evidence that in some patients, pre-existing tics and TS may be a predisposing factor for the development of rapid-onset functional tic-like behaviors [65,66,67,68].

Some patients recapitulate features encountered in a skewed social media representation of tics. In three studies, expert tic clinicians reviewed popular social media videos seen by many of the abrupt-onset cases, and concluded that “TS symptom portrayals on highly viewed TikTok videos are predominantly not representative or typical of TS” [7,69]. Portrayals were heavily skewed for environmentally responsive, aggressive, self-injurious or throwing behaviors and coprolalic utterances. Not surprisingly, therefore, tic-like behaviors developing “after social media consumption differ from tics in Tourette’s syndrome, strongly suggesting that these phenomena are categorically different conditions” [18].

However, the quality of the evidence for which clinical features can accurately differentiate functional tic-like symptoms from TS varies, and some features are less useful in diagnosing individual patients, as discussed in the next sections.

## 5. How to Differentiate Tics and Functional Tic-Like Symptoms

### 5.1. Quality of Evidence

Early in the upsurge of cases in the past few years, many clinicians noted marked differences between them and the perhaps hundreds of tic patients they had evaluated previously. Such evidence should not be ignored, but potentially is vulnerable to bias and susceptible to differences in exposure of the clinician. Later reports compared specific features between groups of typical and FND tic patients, sometimes with prospective ascertainment. However, in many cases the diagnostic approach used to divide the groups presumably used some of the same features, such that the results may be criticized as possibly circular. For instance, if clinicians expected to see more FND in female patients, and explicitly or implicitly included sex in their clinical diagnostic approach, they would potentially classify more females in the clinical FND groups. Of note, there has not been a clear and consistent definition of functional tics used consistently across studies, and expert clinician impression is typically relied upon.

Fortunately, some more recent studies have incorporated strategies to address these limitations. For instance, in some such studies, expert clinicians blind to clinical diagnosis reviewed symptoms in groups identified by other clinicians [18]. Other studies prospectively ascertained clinical features of clinical groups defined by a single criterion chosen a priori (e.g., rapid onset) [5,8,70].

### 5.2. Summary of Evidence

Appendix A summarizes published reports on the phenomenology of functional tic-like behaviors, and includes a column for “Evidence type” so the reader can identify which strategies were utilized in the different reports [12,20,68,71,72,73,74,75,76,77,78]. Specific clinical features are grouped so that existing support for each feature can be reviewed together.

## 6. Recommendations

### 6.1. Assessment

Recent reviews discuss recommendations for clinical assessment and quantification of motor FNDs [53,79] (T. R. Nicholson et al., 2020; Perez, Aybek, et al., 2021). Thorough assessment, including attention to the biopsychosocial model, is the first step in clarifying the presenting diagnosis. The clinician should ask the patient about symptom phenomenology, onset, and progression. The patient should also be screened for co-occurring disorders, including depression, anxiety, and adjustment disorders, asked about a history of any other FND, and asked about any recent stressors, including traumas (though it is important to note a traumatic event is not needed in order to have symptoms consistent with FND). The patient should be asked about exposure to “Tourette” videos on social media or other internet sites; however, this too should be taken within the greater context of symptoms. In addition, one should ask about the patient’s and family’s history of tics (even if subtle, or previously subsided) and of commonly co-occurring disorders, including ADHD and OCD. Finally, outside of the particular tic-like symptoms themselves, it is important to assess the functional impairment incurred from the symptoms, including the impact on other physical and psychological factors, including quality of life, as this information can help to inform treatment [79].

### 6.2. Diagnosis

#### Varying Diagnostic Utility of Clinical Features

The list in Table 1 takes into account the varying quality of evidence listed above, and suggests features common to functional tic-like behaviors that may help to distinguish FND from TS when taken together in a comprehensive clinical evaluation. However, most of these features are of little use in isolation. At the top of the list appear features that differ significantly in prevalence within FND vs. typical tic disorders, but that are not very useful in isolation with regard to differential diagnosis. For instance, anxiety is more common in FND than in TS, but it is common enough in TS that its presence gives little differential diagnostic information [76]. At the bottom of the list appear features that individually produce a high posterior probability of FND. For instance, coprolalic utterances at onset are very common in FND-related tic presentations, while specific complex vocal tics such as coprolalia are quite rare at the onset of typical tic disorders. Importantly, the clinician rarely has to rely on a single feature to make a diagnosis. Thus when formulating whether the symptoms are more consistent with a primary tic disorder, a functional movement disorder or both, it is important to understand the complexity of symptoms as a whole and in the overall context of the presentation. Similarly, it is helpful to evaluate the symptoms over a period of time rather than using a clinical snapshot to make the determination [63]. The combination of numerous features from Table 1 makes the diagnosis more confident; an example might be abrupt onset of severe tic-like movements in the limbs, plus ten different non-suppressible, long, socially inappropriate sentences, in a 17-year-old girl with no prior tics, after social media exposure to identical symptoms. The entire constellation of symptoms, time course, and context are considered in approaching diagnosis.

Nevertheless, we acknowledge that some individual cases may be difficult to classify, and in such cases, diagnostic humility is justified pending further information [63,72,80]. Examples may include patients with typical TS in childhood, then few tics for 5 years, followed in late adolescence by an abrupt recrudescence of severe tics with some unusual features. Some such patients may have TS, while others may have both TS and FND.

## 7. Management

### 7.1. Avoiding Risk

Delivering an accurate diagnosis of FND is critical to avoid the risk of unneeded medical interventions. Specifically, antipsychotic and immunomodulatory therapies have no role in managing FND, unless other diagnoses require them. Brain imaging, electroencephalogram (EEG), and blood work are usually not necessary. They can be harmful as they may introduce ambiguity and confusion regarding the diagnosis and may inadvertently reinforce the state of perceived illness. However, if specific clinical features merit work-up, the clinician may judiciously order investigations in a targeted fashion.

### 7.2. Treatment

The optimal multidisciplinary treatment for FND has not been standardized; recent reviews include the following references [42,81,82,83].

Most experts believe that with early diagnosis and intervention, the prognosis is good and full recovery is possible [81]. Appropriate counseling about the nature of the diagnosis is key. Although we must acknowledge that some individuals may manifest both TS exacerbations and FND, differentiating the treatment of tics and TS from that of FND is important. In individuals who have relapsing or persistent symptoms, a multi-disciplinary approach is likely the best strategy.

### 7.3. Education

The first priority in managing FND is education. This includes making a confident, specific diagnosis using the words “functional neurologic disorder” when the diagnosis is clear. It is important to explain that FND can be completely distinct from TS, although the outward features may be similar. An article by Carson et al. (2016) provides helpful and practical tips for communicating the diagnosis effectively, and a video describes for clinicians a neuroscience-based approach to delivering an FND diagnosis [84,85,86]. Explaining how the diagnosis is made and why potentially harmful additional tests are not needed is often necessary. The patients should be counseled that while medications are generally not helpful in this condition (aside from treating comorbid mood or anxiety disorders), treatment with a combination of psychology and other therapy services often is very effective at reducing and/or completely resolving symptoms.

Individuals and their supporting caregivers or partners should be educated and offered resources to learn more. Specific resources that can be helpful are neurosymptoms.org (regarding FND in general) and its content specific to functional tics (https://www.neurosymptoms.org/en_GB/symptoms/fnd-symptoms/functional-tics/, accessed on 21 October 2022), the TAA statement that addresses this topic in lay language (https://tourette.org/rising-incidence-of-functional-tic-like-behaviors/, accessed on 21 October 2022), and a website that links several resources specifically for the current epidemic of functional tic-like behaviors (https://cumming.ucalgary.ca/resource/tourette-ocd/children-and-adults/disorder-specific-resources/tourette-syndrome-and-0) (accessed on 21 October 2022).

### 7.4. Trigger Reduction

There are frequently personal or environmental influences that can increase the expression of functional tic-like behaviors. While some may be predictable across many people with FND, others are personal and individual factors. Thus it is critical for a clinician to help identify (and ultimately reduce) exacerbating factors.

Some models for understanding rapid onset functional tic-like behaviors suggest that predisposing traits may include genetic factors, early life events, introspective awareness, and social cognitive traits (for further discussion, see the following references: [5,23]. Acute contributions can include psychosocial stressors (e.g., anxiety, and depressed mood), social isolation, and life stressors (e.g., abrupt transitions in educational programs and/or academic demands). Exposure to tic-like behaviors in this context may lead to developing those same behaviors. Increased attention following complex tic-like behaviors and related avoidance of stressful circumstances (such as school) may inadvertently reinforce and increase the frequency of these symptoms. Considering this model, identifying triggers that may provoke FND manifestations is a critical step.

For some young people, engaging with social media content related to tic manifestation, such as watching videos of highly visible personalities with tics and tic-like behaviors, may provoke similar manifestations [18,87]. This phenomenon shares some similarity with the well-known experience in primary tic disorders that discussing or thinking about tics may increase the likelihood they occur [64,88]. More time watching social media was significantly associated with greater tic severity and lower quality of life, even though 95% of the teens with tics surveyed did not search for tic content online [19]. In functional tic-like behaviors, the role of social media has been suspected to be especially impactful.

Understandably, social media has become a cornerstone of social interaction for many young people. As such, many young people may interact with social media content that can influence the expression of tics and/or tic-like behaviors. In cases in which social media content is exacerbating and/or worsening tics and/or tic-like behaviors, it would be prudent to reduce access to social media (along with other factors that exacerbate symptom expression) for a period of time. As patients learn skills to effectively manage tics and tic-like behaviors in treatment, access could be reintroduced in a graduated manner [23].

Personal stressors can be critical to explore. Research in non-epileptic seizures has shown an increased incidence of past abuse as compared with epilepsy patients, and adverse life events have been proposed as a potential contributing factor to the development of FND [89]. It is important to emphasize that there is often a delay between adverse events and the development of functional symptoms [23]. Stressors may include sleep deprivation, in which case addressing sleep hygiene and routines is advisable [6]. Furthermore, if significant stressors are not apparent, it is important to emphasize that ongoing “normal” stresses of life may contribute. Physiological arousal, including disrupted sleep or signs of anxiety, may support such a conclusion [6]. Counseling to explore the impact of life circumstances and to learn to recognize and manage stress is critical, and may be one of the most important aspects of management. It should be noted that not all FND cases have stress or adverse events as a contributing factor, and the absence of an obvious trigger should not preclude the diagnosis. 

### 7.5. Reducing Reinforcement

The occurrence of functional tic-like behaviors often has an impact on school, personal, or professional life. It is important to balance nurturing support and empathy, which are appropriate, against inadvertently reinforcing the symptoms by allowing avoidance of stressful but necessary aspects of life. The goal should be to develop skills to cope with the functional tic-like behaviors while simultaneously working to reduce or eliminate the symptoms.

While these functional tic-like symptoms, like other FND symptoms, are not intentional, there may be some unintentional reinforcement or secondary gain in some patients. Exploring that possibility can help identify possible interventions to reduce the symptoms [20].

### 7.6. Comorbid Symptoms and Disorders

When a person with functional tic-like behaviors also suffers from additional conditions, such as anxiety, depression or insomnia, it is critical to identify and address those co-occurring conditions through behavioral therapy and/or medication. Addressing these important contributing factors makes resolving functional disorders more likely.

### 7.7. Retraining the Brain

It is often helpful to describe the process of treating the symptoms as “retraining the brain.” In addition to psychology to help identify possible triggers, this process can be augmented with physical, occupational and/or speech therapy [90,91,92]. While the specific treatment regimens will differ depending on the patient and specific abnormal movements or sounds, the overall goals should be to reinforce normal motor patterns and redirect the unwanted movements or sounds. A key is to reestablish a sense of control over these movements and sounds.

While Comprehensive Behavioral Intervention for Tics (CBIT) was developed and validated for the treatment of tics in TS, and has not been extensively studied in functional tic-like behaviors, there have been some reports that a modified CBIT program can be helpful. Proposed modifications include focusing more on functional behavioral assessment components, such as exploring the internal and external antecedents of the tic-like behaviors and exploring the consequences that the behaviors induced [8,23].

While studies specifically on functional tic-like behaviors are not available, there is good reason to think that Cognitive Behavioral Therapy (CBT) or psychodynamic psychotherapy may also help [93,94,95]. It has been proposed that motivational interviewing and CBT can help reduce stress, address anxiety and comorbidities, and improve adaptive functioning [23,83,96]. Evidence for pharmacotherapy in the absence of co-occurring conditions is limited [97,98,99].

### 7.8. Treatment: Summary

In summary, treatment must be individualized. A treatment strategy should ideally include education on the nature of FND, avoiding interventions and treatments not indicated, and avoiding triggers as well as exacerbating factors. CBT for FND may be recommended and helpful. Careful avoidance of reinforcing the symptoms is critical. CBIT offers a well-studied model to reduce tics, reduce reinforcement of tics and tic-like behaviors, and may be useful for other functional movement symptoms. Addressing depression or anxiety, when present, and optimizing sleep comprise an additional and important component of care.

## 8. Recommendations for Future Studies

Functional neurological disorders are complex, and may be entangled with typical manifestations of neurological and other disorders. Future research is needed to determine optimal management for these difficult disorders. Specific areas recommended for further study include:Measuring consistency of features across sitesFurther delineation of sex effects on TS and on FND-ticsExploring the influence of reinforcing environmental responsesDevelopment of objective clinical diagnostic measures (e.g., ref. [100]). Ideally, knowing the sensitivity and specificity of particular clinical features such as those in Table 1, together with each feature’s prior probability, might conceivably create a “risk calculator” to estimate a posterior probability of FND, along with global clinical impression.Functional imaging studies, such as fMRI, to establish similarities/differences in patients with different presentations of FNDEpidemiological or multicenter studies on whether FND-tics are increasing or waning at this point in time. Some clinicians have noticed that the FND-like presentations seem to have peaked in frequency, and the CDC’s national data from emergency departments would support that interpretation [13].Outcome studies with different interventions.

The authors advocate for collaborative work to address these “need” areas to ensure identification and optimization in management.

## Figures and Tables

**Table 1 jcm-11-06470-t001:** Features of relevance to differentiating FND from TS.

*Features near the top of the list are more common in FND than in TS, but are seen in TS as well, and are not diagnostic in isolation. Features may not be unique to either condition, nor prerequisite, but are more commonly present. Features near the bottom of the list strongly suggest FND.*
Anxiety (higher prevalence) Female predominance (greater proportion of affected individuals) No family history of tics, OCD or ADHD No personal history of OCD, ADHD, or earlier typical tic disorder No urge to tic or premonitory sensation (some young children do not report urges or premonitory sensations) Onset in the mid-teen years Significant variability of tic symptoms (vs. stereotyped tics) No response, or atypical response, to proven tic-suppressing medications Symptoms in extremities before face and neck (lack of typical rostrocaudal progression) Symptoms that dramatically and persistently disrupt the person’s intended actions or communications Symptoms that the patient has not previously experienced that closely replicate those of someone whom the patient has observed Extreme “attacks” of tic-like behavior—discrete episodes of complex movements and/or vocalizations, lasting from minutes to several hours with an abrupt onset and offset, often described as appearing to be “seizure-like” and often early in the course (Robinson and Hedderly, 2016 and Heyman et al. 2021) Inability to suppress, even temporarily Severity of movements and vocalizations is constant over time rather than waxing and waning Movements or vocalizations that are dramatically more severe in the presence of others versus when alone Onset in adulthood Presence of other functional neurological symptoms Sudden, abrupt onset Severe symptoms at onset Coprophenomena at onset Lack of perceived agency (a sense that the movement is done to the patient rather than by them) Tics involving the body or limbs without a history of tics involving the eyes, face, and head

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
