# Peer review of "Diagnosis and Management of Functional Tic-Like Phenomena"

_jcm, 2022, doi:10.3390/jcm11216470_

Round 1

Reviewer 1 Report

Detailed remarks:

Lines: 69-70 Baizabal-Carvallo 69 & Jankovic, 2014 – this article does not come from years 2020-2021 as pointed in line 69

Lines: 100-104. There is some evidence that suggests tics in females may be more complex, may begin later in life, and may be more often complicated by mood and anxiety disorders (Garcia-Delgar et al., 2021; Garris & Quigg, 2021), but conflicting data also exist, and these points require  further investigation. Regarding comorbidity these conflicting date were not cited e.g. Freeman et al. 2000 (this article was already cited in manuscript) found that externalizing behavior and ASD were less often seen in female than in males, and mood and anxiety disorders were similarly seen in both sexes.

Would you please give information on definition of functional tics the authors of publications you cite used? I am afraid that different clinicians include various hyperkinesis/behaviors/motor activity into the term functional tics and this may have impact on inconsistent results generally. Look at your Table S1 (otherwise tremendous). We can find terms: myoclonic jerks, seizure-like behavior, rhythmic movements (rhythmic is tremor, not tics), throwing objects etc. It is very difficult to distinguish tics from myoclonus when vocalizations are absent? So, there is great variety of behaviors that are labelled as functional tics. In conclusion, can you define functional tics?

Can you give information about total number of patients who were included in the studies you cited in S1 Table. How big is the group of functional tic patients ?

Main table: Features differentiating FND from TS. Generally I agree but let me analyze some of them and give polemic statements. First sentence comes from the table, second one is from me.

·        Anxiety – appear in about 1/3 of TS patients, is often related to hyperactivity/motor anxiety or avoidance which could be considered as some clinician as tic-like phenomenon (vide: considerations on functional tics definition); anxiety-reducing medication sometimes  improve organic tics/tic-like movements (not only functional tics).

·        Female predominance – ¼ of  TS patients are females

·        No family history of tics, ADHD, OCD – I agree but in case of coexistence primary tic disorders with tic-like functional movements the family history could be positive.

·        No personal history of OCD, ADHD, or earlier typical tic disorder – look at your own Table S1 where you find opposite examples, contrary to above statement.

·        No premonitory urges – the authors themselves provided us that this is not always true.

·        Onset in mid-teen years – according to DSM-5 TS may start up to 18

·        No response to tic reducing medication – what about refractory TS that is well known in clinical practice.

·        Extreme attacks of tic-like behavior – organic tics may appear in bouts/series of attack.

·        Onset in adulthood – of course I agree but pay attention to secondary tics such as drug-induced, neurodegenerative disorders (e.g. neuroacanthocytosis, Wilson’s)

·        Sudden abrupt onset –  according to Freeman R (2000) 4.9% of TS patients started abruptly.

·        Description of an urge very different in kind and duration compared to typical premonitory urges in TS – unclear, give an example what you mean in text

Would you provide the reader with comments on the above disputatious statements, at least some of them? Other features listed in table are not questionable.   

 I do not understand the title of Table. FND compromise many functional symptoms, much more than even functional movement disorders (weakness, ataxia etc). This table concerns only functional tics, not FND. I have impression that these two terms, FND and functional tic-like behavior, are used by authors also in text alternatively, as synonyms, which in my opinion is not justified and not true.

Treatment. In some functional movement disorders low dose, below threshold to give benefit, medications are used e.g. botulinum toxin in patients with functional focal dystonia. Would you comment this strategy with regard to functional tics? Are there any data in this field?

Finally, it should be strongly pointed that there are not one or two features that enable diagnosis of functional tics. Only combination of many features taken together makes the diagnosis appropriate. This statement appears in text (line 276) but should be listed in separate paragraph, as the one of main conclusions of the study. 

Reviewer 2 Report

The Authors in this review describes the presentation of the emerging phenomenon of Functional Neurological Disorder. In the manuscript the diagnosis, the symptoms, the relationship with TS and the management have been discussed largely.

The literature has been reviewed too, from the very beginning (first reports of this phenomenon) to the very last papers. In supplementary materials, a table with data for features of functional vs typical tics should depict the clinical tic's feature of FND vs TS. However, the table is not clear in some parts (eg in Radhakrishnan et al the statistics column seem to be hard to understand and should be better described).

The recommendation chapter is on of the most valuable of the manuscript: it provides clues in assessment, diagnosis providing also a table differentiating FND an TS features. 

The management chapter provides a useful tool for the treatment.

Overall, this manuscript offer a wide and comprehensive point of view of FND.

Author Response

Thank you for identifying the opportunity to improve clarity.  We have made edits throughout the table to improve this.